# High Glucose Impairs Expression and Activation of MerTK in ARPE-19 Cells

**DOI:** 10.3390/ijms23031144

**Published:** 2022-01-20

**Authors:** Alessandra Puddu, Silvia Ravera, Isabella Panfoli, Nadia Bertola, Davide Maggi

**Affiliations:** 1Department of Internal Medicine and Medical Specialties, University of Genova, 16132 Genova, Italy; davide.maggi@unige.it; 2Dipartimento di Medicina Sperimentale, Università di Genoa, Via De Toni 14, 16132 Genova, Italy; silvia.ravera@unige.it (S.R.); nadia.bertola@gmail.com (N.B.); 3Dipartimento di Farmacia (DIFAR), Università di Genova, V.le Benedetto XV 3, 16132 Genova, Italy; isabella.panfoli@unige.it

**Keywords:** MerTK, ARPE-19 cells, diabetes, ADAM9, miR-126, phagocytosis

## Abstract

MerTK (Mer Tyrosine Kinase) is a cell surface receptor that regulates phagocytosis of photoreceptor outer segments (POS) in retinal pigment epithelial (RPE) cells. POS phagocytosis is impaired in several pathologies, including diabetes. In this study, we investigate whether hyperglycemic conditions may affect MerTK expression and activation in ARPE-19 cells, a retinal pigment epithelial cellular model. ARPE-19 cells were cultured in standard (CTR) or high-glucose (HG) medium for 24 h. Then, we analyzed: mRNA levels and protein expression of MerTK and ADAM9, a protease that cleaves the extracellular region of MerTK; the amount of cleaved Mer (sMer); and the ability of GAS6, a MerTK ligand, to induce MerTK phosphorylation. Since HG reduces miR-126 levels, and ADAM9 is a target of miR-126, ARPE-19 cells were transfected with miR-126 inhibitor or mimic; then, we evaluated ADAM9 expression, sMer, and POS phagocytosis. We found that HG reduced expression and activation of MerTK. Contextually, HG increased expression of ADAM9 and the amount of sMer. Overexpression of miR-126 reduced levels of sMer and improved phagocytosis in ARPE-19 cells cultured with HG. In this study, we demonstrate that HG compromises MerTK expression and activation in ARPE-19 cells. Our results suggest that HG up-regulates ADAM9 expression, leading to increased shedding of MerTK. The consequent rise in sMer coupled to reduced expression of MerTK impairs binding and internalization of POS in ARPE-19 cells.

## 1. Introduction

The retinal pigment epithelium (RPE) is formed by a monolayer of highly specialized cells, with the basolateral side facing Bruch’s membrane and the choroid, and the apical side in contact with the outer segments of photoreceptor cells [1]. Functions of RPE cells are essential for visual function and contribute to maintain homeostasis of the retina [2,3]. In particular, RPE is involved in the phagocytosis and digestion of photoreceptor outer segments (POS), which is of particular importance for photoreceptor function and survival. Indeed, failure of POS uptake leads to progressive degeneration of photoreceptor cells, thus compromising the integrity of the retina [4].

The process through which RPE cells phagocytize POS belongs to a conserved mechanism, comparable to that employed by phagocytes to remove apoptotic cells. Phagocytosis of POS is finely regulated by Mer Tyrosine Kinase (MerTK), a cell surface receptor member of the tyro/Axl/Mer family of receptor tyrosine kinase [5]. On the one hand, MerTK undergoes autophosphorylation after binding its ligands, Growth arrest-specific 6 (Gas6), or Protein S, thus activating intracellular signaling, which leads to POS engulfment [2,4,6]. On the other hand, the extracellular region of MerTK may be cleaved by Human A Disintegrin And Metalloproteases (ADAMs) and released as a soluble form (sMer) [7,8]. It has been demonstrated that sMer acts as a decoy receptor of MerTK ligands, thus inhibiting MerTK activation and, consequently, phagocytosis. Interestingly, POS phagocytosis follows a circadian rhythm in vivo, with a peak of activity 1.5–2 h after light onset that corresponds with maximum levels of MerTK phosphorylation, whereas levels of sMer increase after phagocytosis [9,10]. Law et al. showed that the presence of sMer in the extracellular space decreased POS binding to the RPE surface, whereas inhibition of MerTK cleavage increased it [9]. Therefore, the balance between the full-length and the soluble form of MerTK may represent a critical regulatory point of POS phagocytosis. Increased shedding of MerTK has been reported in macrophages exposed to high glucose (HG) [7]. Contextually, this study demonstrated that ADAM9 is responsible for MerTK shedding, and that miR-126 regulates ADAM9 expression. Specifically, HG exposure decreased miR-126 levels, with a corresponding increment in ADAM9 expression. By contrast, overexpression of miR-126 can attenuate ADAM9 expression [11].

RPE dysfunction is responsible for several diabetic ocular diseases, often related to retinal–blood barrier breakdown. In addition, phagocytosis of photoreceptors is impaired in several pathologies, including diabetes [2,12,13]. However, the molecular mechanisms underlying the dysfunctional POS phagocytosis under HG conditions remain to be elucidated. We previously demonstrated that exposure of the RPE cell line ARPE-19 to HG significantly decreased miR-126 levels [14]. In this study, we show that HG impairs the expression and activation of MerTK in ARPE-19 cells, leading to defective phagocytosis. Moreover, we found that overexpression of miR-126 can reverse HG damage by down-regulating ADAM9 production.

## 2. Results

### 2.1. High Glucose Decreased MerTK Expression

MerTK regulates POS phagocytosis in the retinal pigment epithelium [2]. Impairment of phagocytosis has been described in many ocular diseases, including diabetic complications [2,12,13]. Therefore, we investigated the effect of HG on the expression of MerTK in ARPE-19 cells.

Hyperglycemic conditions significantly reduced levels of MerTK mRNA (Figure 1A).

Analysis of protein expression of MerTK revealed that expression of MerTK is down-regulated after exposure to HG (Figure 1B,C).

### 2.2. High Glucose Reduced MerTK Activation

To evaluate whether HG also affects MerTK activation, cells were stimulated for 5, 10, and 15 min with 2 μg/mL Gas6, a MerTK ligand. We found that Gas6 induced hyperphosphorylation of MerTK in CTR cells, reaching its maximum after about 15 min (Figure 2). When cells were cultured with HG, Gas6 was not able to induce significant stimulation of MerTK.

Gas6 interacts with the extracellular region of MerTK, leading to receptor activation [15]. To evaluate whether HG affected the amount of the MerTK receptor that could be potentially activated by Gas6, we used an antibody that recognizes the extracellular region of MerTK, and we found a decreased amount of the corresponding band when cells were cultured in HG (Figure 3).

### 2.3. High Glucose Increased ADAM9 Expression

It has been reported that ADAM9 is responsible for MerTK shedding [7]. Therefore, we investigated the effects of HG on ADAM9 expression and production in ARPE-19 cells. RT-qPCR revealed that HG increased mRNA levels of ADAM9 (Figure 4A). ADAM9 is synthesized as an inactive proenzyme with a molecular weight of 100 kDa. The active enzyme (80 kDa) results from proteolytic cleavage of the prodomain [16]. Exposure of ARPE-19 cells to HG significantly increased the amount of both precursor and active form of ADAM9 (Figure 4B,C).

### 2.4. High Glucose Increased MerTK Shedding

To evaluate the activity of ADAM9, the amount of sMer in conditioned media collected after exposure to HG was quantified by ELISA.

The results show that sMer was almost undetectable in supernatants of ARPE-19 cells cultured under control conditions (Figure 5). The amount of sMer released in the medium significantly increased after treatment of ARPE-19 cells with HG.

### 2.5. ADAM9 Is a Target of miR-126 in ARPE-19 Cells

It has been reported that ADAM9 is a target of miR-126 in macrophages [7]. Since we previously demonstrated that HG decreases miR-126 levels in ARPE-19 cells [14], we investigated whether miR-126 affects ADAM 9 expression.

Treatment with miR-126 inhibitor increased mRNA levels of ADAM9 in cells cultured under normal glucose concentration (CTR), as well as in cells exposed to HG (Figure 6A).

Analysis of the protein levels of ADAM9 showed that the proenzyme expression was up-regulated after the addition of miR-126 inhibitor (Figure 6B,C). The same trend was found for the active form (Figure 6D).

On the contrary, transfection of miR-126 mimic reduced mRNA levels of ADAM9 and down-regulated protein expression (Figure 6E–H).

To verify whether miR-126 affects the activity of ADAM9, we evaluated the levels of sMer in ARPE-19 cells transfected with miR-126 inhibitor or mimic. Transfection of ARPE-19 cells with miR-126 inhibitor significantly increased levels of sMer (Figure 7A). Overexpression of miR-126 counteracted the rise of sMer in cells exposed to HG (Figure 7B).

### 2.6. High Glucose Impaired POS Phagocytosis

Finally, since increased levels of sMer decreased phagocytosis [9], we evaluated whether miR-126 may affect POS engulfment. To this aim, we analyzed the levels of rhodopsin, a POS-specific protein, after phagocytosis assay. Our data showed that the amount of rhodopsin detected in lysates from cells cultured under HG conditions was significantly decreased compared to control cells (Figure 8). Transfection with miR-126 mimic had no impact on control cells but increased rhodopsin detection in cells cultured in HG.

## 3. Discussion

Phagocytosis of POS is one of the essential functions of RPE cells and is regulated by the cell surface receptor MerTK [2,4,6]. RPE dysfunction is responsible for several diabetic ocular diseases; however, it has often been related to retinal–blood barrier breakdown, whereas the effects of HG on the phagocytic activity have been poorly investigated. In this study, we demonstrate that HG compromises MerTK expression and activation in ARPE-19 cells.

Firstly, we found that hyperglycemic conditions decreased the expression of MerTK in ARPE-19 cells. In particular, HG affected MerTK at different levels by down-regulating MerTK expression, as indicated by the mRNA transcript decrease, and by inducing the loss of the receptor extracellular region. These results suggest that HG may affect MerTK activation by reducing the amount of receptor that can be activated by its ligands. According to this hypothesis, we found that Gas6-induced MerTK activation was reduced in ARPE-19 cells exposed to HG.

Several studies demonstrate that the activation of MerTK mediated by Gas6 is reduced in the presence of the soluble form of the MerTK receptor. Indeed, sMer, by acting as a decoy receptor for MerTK ligands, prevents their interaction with the receptor, thus antagonizing their action [9,17,18]. Considering that HG reduces detection of the MerTK extracellular domain as well as the ability of Gas6 to stimulate MerTK phosphorylation in ARPE-19 cells, our results suggest that HG may induce MerTK shedding. In our model, HG induced an evident increase in the amount of sMer in the media of ARPE-19 cells, confirming that HG induces MerTK cleavage. Suresh Babu et al. reported that increased MerTK shedding is correlated to up-regulation of ADAM9 expression in macrophage cell lines cultured under diabetic conditions [7]. In accordance with this evidence, we found that HG increases expression of ADAM9, and that this increment is coupled to an increase in the amount of sMer in the media of ARPE-19 cells cultured under hyperglycemic conditions, suggesting that ADAM9 is also involved in MerTK cleavage in this cell line. Furthermore, Suresh Babu et al. demonstrated that up-regulation of ADAM9 expression is mediated by reduced levels of miR-126 [7]. Interestingly, we previously demonstrated that HG reduces miR-126 levels in ARPE-19 cells [14]. Here, we found that inhibition or overexpression of miR-126 leads to up-regulation or down-regulation, respectively, of ADAM9 expression, demonstrating that ADAM9 is a target of miR-126 in ARPE-19 cells. Contextually, we observed that inhibition of miR-126 increased the levels of sMer in ARPE-19 cells, whereas miR-126 overexpression strongly reduced the amount of sMer in media from cells cultured under diabetic conditions. Lack of significant reduction in sMer levels in ARPE-19 cells treated with miR-126 mimic under normal glucose conditions may be due to the low levels of sMer found in the medium collected from CTR cells. Our data demonstrate that the rise in sMer detection is correlated with the increase in ADAM9 expression, confirming that ADAM9 is involved in MerTK shedding. Here, we provide evidence that miR-126 plays a role in HG-induced cleavage of MerTK. Therefore, HG, by altering miR-126 levels, may regulate MerTK function, affecting ADAM9-mediated shedding.

Elevated levels of sMer impair the engulfment of apoptotic cells by macrophage cultured under hyperglycemic conditions [18]. Phagocytosis of POS is one of the important functions of RPE cells and shares several analogies with efferocytosis, including the involvement of MerTK. In particular, MerTK plays a crucial role in POS phagocytosis: it is necessary for POS engulfment, and, at the same time, it regulates the rate of phagocytosis. Indeed, it has been reported that MerTK-deficient RPE cells are not able to engulf POS, leading to photoreceptor degeneration [3]. Moreover, MerTK cleavage contributes to the acute regulation of RPE phagocytosis by limiting POS binding to the cell surface [9]. Further, the increased release of sMer inhibits Gas6 activity, thus blocking signaling pathways that lead to POS internalization. These activities depend on the rate of MerTK cleavage and on the decoy propriety of sMer. Indeed, blocking the cleavage of MerTK increases the amount of POS that can be bound at the surface, whereas the rise in sMer, acting as a decoy receptor, reduces the effects of MerTK ligands, leading to decreased POS binding [9]. Therefore, under physiological conditions, sMer may be considered a brake for phagocytosis, intended to avoid an excessive POS load. On the other hand, by decreasing POS binding and MerTK activation, the rise in sMer may slow down RPE function, leading to pathological accumulation of shed POS. Our results suggest that the latter scenario probably represents what occurs under hyperglycemic conditions *in vivo*. Indeed, we found that the amount of rhodopsin decreased in lysates from cells cultured with HG, suggesting that the increased amount of sMer may impair POS engulfment in ARPE-19 cells. Interestingly, overexpression of miR-126 in cells exposed to HG restores the amount of rhodopsin to the levels observed in ARPE-19 cells cultured under normoglycemic conditions, confirming that POS engulfment is directly correlated with levels of sMer.

Increased thickness of the RPE–photoreceptor complex has been found in patients with type 2 diabetes [19,20]. Our results support the hypothesis that this alteration may be due to the accumulation of shed POS caused by phagocytic RPE dysfunction. Moreover, since lesions in the RPE-photoreceptor complex are also found in diabetic subjects without diabetic retinopathy [20], our results suggest that phagocytic dysfunction precedes diabetes-induced ocular microvascular alteration. Recent studies also demonstrate that neuronal alterations also precede microvascular damage in diabetic retinopathy [21].

The mechanisms through which hyperglycemia affects POS phagocytosis are not yet understood. Our study suggests that the HG-induced decrease in levels of miR-126 leads to up-regulation of ADAM9 expression, and this, in turn, increases cleavage of MerTK. The consequent rise in sMer coupled to reduced expression of MerTK may impair activation of intracellular signaling that leads to binding and internalization of POS (Figure 9). Moreover, we provide evidence that hyperglycemia affects POS phagocytosis, highlighting the importance of preserving RPE function.

In conclusion, considering that soluble MerTK receptor promotes endothelial cell recruitment in breast cancer [17] and that choroidal neovascularization is reduced in ADAM9 k/o mice, it may be hypothesized that increased levels of sMer induced by HG, besides impairing phagocytosis, may also favor ocular neovascularization. In light of this evidence, it is necessary to take into account the fine balances that govern retinal homeostasis in the choice of strategies to counteract retinal degeneration.

## 4. Materials and Methods

### 4.1. Cell Culture and Experimental Conditions

The human cell line ARPE-19 (American Type Culture Collection, Manassas, VA, USA) from passages 22 to 28 were grown in DMEM/F12 1:1 medium (Euroclone, Milan, Italy) supplemented with 10% fetal bovine serum and 2 mmol/L glutamine (Euroclone, Milan, Italy) at 37 °C in 5% CO_2_. The cell medium was replaced every 2 days. Cells were grown to confluence, removed with trypsin-EDTA (Euroclone, Milan, Italy), and then seeded in multiwell plates for all experiments. Before each experiment, cells were washed with phosphate-buffered saline (PBS) (Euroclone, Milan, Italy) and cultured under normal glucose conditions (1 g/L glucose, CTR). Once they reached confluence, cells were cultured in control medium (CTR) or high glucose (4.5 g/L glucose, HG) for 24 h and processed for each analysis.

### 4.2. Quantitative RT-PCR

RNA was extracted from ARPE-19 cells cultured using an RNA/DNA/Protein Purification Plus Kit (Norgen Biotek Corp., Thorold, ON, Canada) according to the manufacturer’s instructions. The amount and quality of the RNA were determined spectrophotometrically. RNA samples with A260/A280 values of at least 2.0 were used for further analysis. Reverse-transcription of 1 µg of RNA to cDNA was performed using Wonder RT-cDNA Synthesis kit (Euroclone, Milan, Italy). Expression levels of the target genes MerTK (Applied Biosystems, Monza, Italy, assay ID: Hs01031968_m1) and ADAM9 (Applied Biosystems, Monza, Italy, Hs00177638_m1) were measured by qRT-PCR amplification, performed using Luna Universal Probe qPCR Master Mix (New England Biolabs, NEB, Ipswich, MA, USA) in an ABI PRISM 7900 HT Fast Real Time PCR System (Applied Biosystems, Monza, Italy). All reactions were performed in triplicate with the following qRT-PCR run protocol: initial denaturation (95 °C, 1 min), denaturation (95 °C, 15 s), and extension (60 °C, 30 s) repeated 43 times (95 °C, 15 s and 60 °C, 1 min). Gene expression was normalized using β-Actin and GAPDH as control genes (β-Actin assay ID: Hs01060665_g1, GAPDH assay ID: Hs02758991_g1, Applied Biosystems, Monza, Italy). Comparisons of gene expression were done using the 2^−ΔΔCt^ method [22].

### 4.3. Immunoblot

At the end of the experiments, ARPE-19 cells were lysed using an RNA/DNA/Protein Purification Plus Kit (Norgen Biotek Corp., Thorold, ON, Canada) or in RIPA buffer supplemented with protease and phosphatase inhibitors. Protein concentrations were determined using a BCA Protein Assay Kit. Fifteen micrograms of total cell lysate were separated by SDS–PAGE and transferred onto nitrocellulose. Filters were blocked in Protein-Free T20 Blocking Buffer (Pierce Biotechnology, Rockford, IL, USA) and incubated overnight at 4 °C with the following primary specific antibodies diluted 1:1000: Mer (D21F11, cat. 4319), Mer (348E6, cat. 9178), and ADAM9 (cat. 2099) from Cell Signaling Technology, Beverly, MA, USA; anti-MerTK (phospho Y749) + TYRO3 (phospho Y681) antibody (cat. ab192649) from Abcam, Cambridge, UK. Secondary specific horseradish peroxidase-linked antibodies, diluted 1:3000, were added for 1 h at room temperature. Bound antibodies were detected using the enhanced chemiluminescence lighting system (LiteAblot EXTEND, Euroclone, Milan, Italy) according to the manufacturer’s instructions. Each membrane was stripped (Restore PLUS Western Blot Stripping Buffer, Pierce Biotechnology, Rockford, IL, USA) and probed for β-actin (1:3000, Cell Signaling Technology, Beverly, MA, USA) to verify equal protein loading. Bands of interest were quantified by densitometry using Alliance software. The results are expressed as percentages of CTR (defined as 100%).

### 4.4. MerTK Activation

To evaluate Gas6-induced MerTK activation, ARPE-19 cells were cultured in standard medium (CTR) or high glucose (HG) for 24 h, and then incubated in serum-free medium for 4 h and exposed for 5, 10, and 15 min to 2 μg/mL to Gas6 (Recombinant Human Gas6 Protein, R&D Systems, Minneapolis, MN, USA). Then, the cells were lysed in RIPA buffer, and phosphorylation of MerTK was evaluated by Western blot using specific antibodies.

### 4.5. Quantification of sMerTK Release

ARPE-19 cells were treated with miRNA mimic or miRNA inhibitor for 24 h. At the end of the incubation, fresh CTR or HG medium were added for 24 h. The conditioned media were collected to quantify sMerTK and stored at −80 °C until the assay was performed. The cells were then washed with PBS, lysed in radioimmunoprecipitation assay (RIPA) buffer, and the protein content was determined with a BCA Protein Assay Kit (Pierce, Rockford, IL, USA) according to the manufacturer’s instructions. Quantification of sMerTK was assessed by enzyme-linked immunosorbent assay (ELISA; RayBiotech, Norcross, GA, USA), and concentrations were calculated from the standards curve and normalized to the total protein concentration of the respective lysate.

### 4.6. miR-126 Mimics/Inhibitor Transfection

ARPE-19 cells were seeded into 12-well plates with 8 × 10^4^ cells/well and cultured under normal glucose conditions. Once the cells reached 70% confluence, they were transfected with has-miR-126-3p miRCURY LNA Power inhibitor (miR-126 inhib: GCATTATTACTCACGGTACG), has-miR-126-3p miRCURY LNA Mimic (miR-126 mimic: UCGUACCGUGAGUAAUAAUGCG), and the respective scramble negative controls (inhib: TAACACGTCTATACGCCCA; mimic: GAUGGCAUUCGAUCAGUUCUA all from Exiqon-Qiagen, Milan, Italy) using jetPRIME transfection reagent (Polyplus-transfection, New York, NY, USA). At 24 h after transfection, the medium was changed and replaced with fresh CTR or high glucose (HG, 25 mM glucose) medium for 24 h. Then, an RNA/DNA/Protein Purification Plus Kit (Norgen Biotek Corp., Thorold, ON, Canada) was used to sequentially purify RNA and proteins from the same sample.

### 4.7. Phagocytosis Assays

Purified bovine POS were prepared as previously described [23]. ARPE-19 cells, previously treated with miRNA mimic or miRNA inhibitor, were pulsed with 8 μg POS in CTR or HG medium; two batches of control cells (CTR and HG) were incubated in DMEM w/o POS. Media containing noningested POS were collected after 4 h at 37 °C, and the cells were extensively washed with PBS to remove unbound POS. Media containing nondigested POS were centrifuged, and the supernatants were stored at −80 °C until used for detection of sMer. Cells were immediately lysed in RIPA buffer supplemented with protease and phosphatase inhibitor cocktails. The protein concentration of each sample was determined using a BCA protein assay Kit.

### 4.8. Statistical Analysis

The results are representative of at least 3 experiments. All analyses were carried out with GraphPad Prism 4.0 software (GraphPad Software, San Diego, CA, USA). Quantifications were expressed as the mean ± SD, and difference between 2 experimental groups was analyzed by Student’s *t*-test. Data of more groups were expressed as the mean ± SD and then analyzed using one-way ANOVA followed by Bonferroni’s multiple comparison test. A *p* value *<* 0.05 was considered statistically significant.

## Figures and Tables

**Figure 1 ijms-23-01144-f001:**
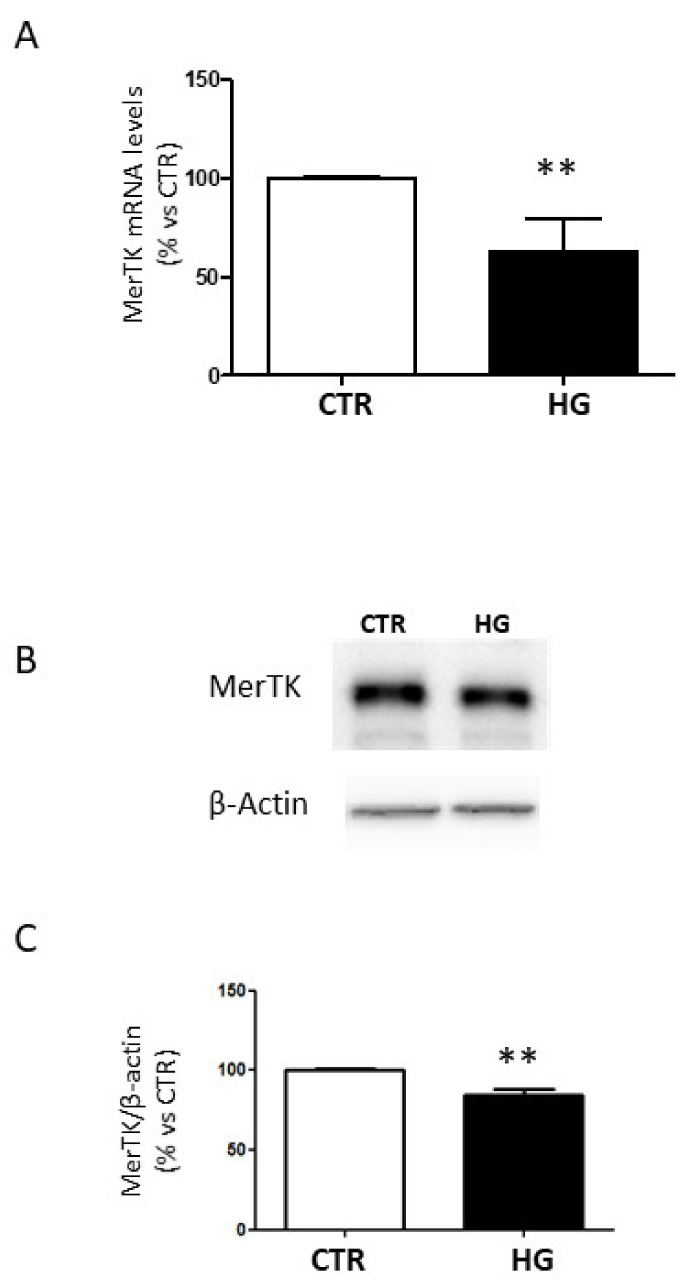
Expression of MerTK in ARPE-19 cells cultured in standard medium (CTR) or high glucose (HG) for 24 h. (**A**) MerTK gene expression normalized vs. housekeeping genes. (**B**) Representative Western blot analysis of MerTK protein expression and (**C**) quantification of densitometries of relative bands. Data are expressed as mean ± SD of fold induction relative to β-actin of three independent experiments (*n* = 3). ** *p* < 0.01 vs. CTR.

**Figure 2 ijms-23-01144-f002:**
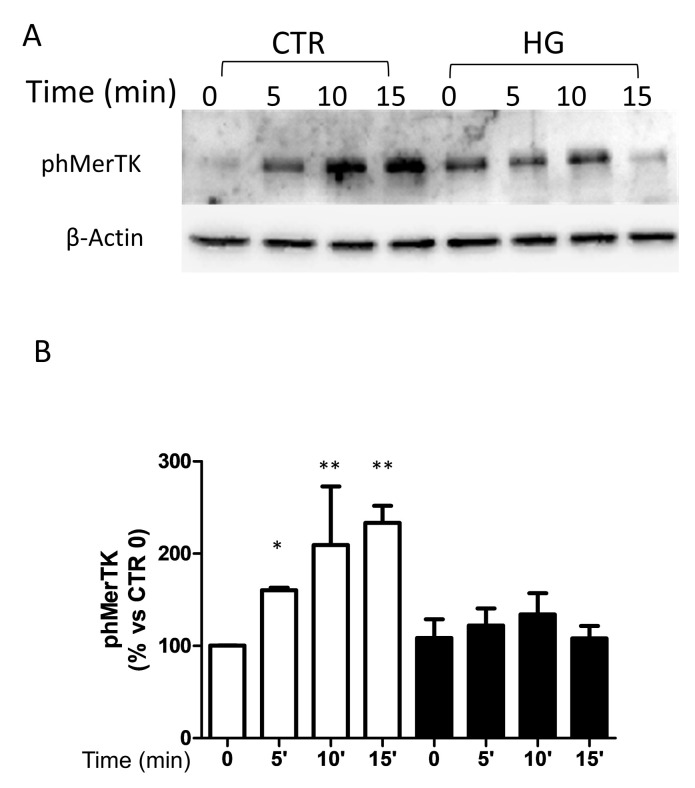
Gas6-induced MerTK phosphorylation. ARPE-19 cells were cultured in standard medium (CTR) or high glucose (HG) for 24 h and then exposed to GAS6 for different times. (**A**) Representative Western blot. (**B**) Quantification of densitometries of relative bands. Data are expressed as mean ± SD of fold induction relative to β-actin of three independent experiments (*n* = 3). * *p* < 0.05 and ** *p* < 0.01 vs. CTR.

**Figure 3 ijms-23-01144-f003:**
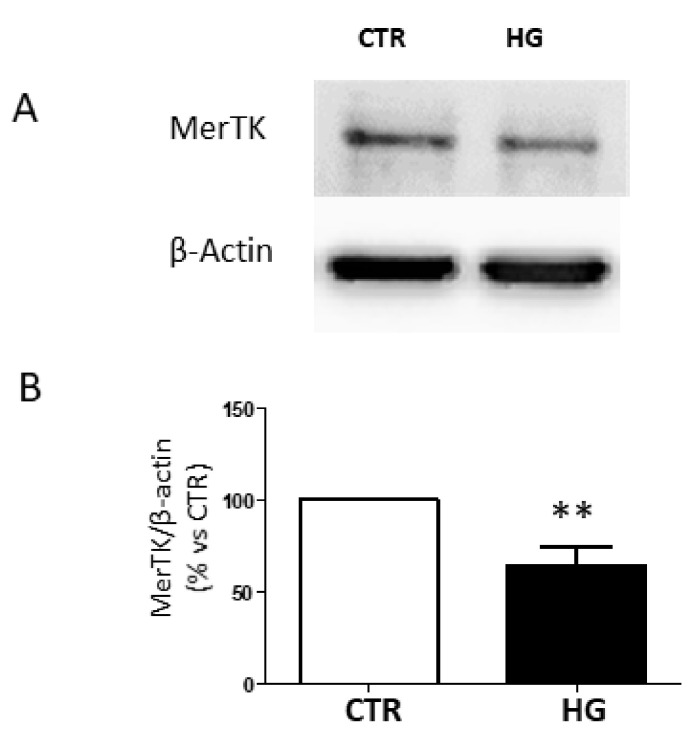
Detection of the extracellular region of MerTK in ARPE-19 cells cultured in standard medium (CTR) or high glucose (HG) for 24 h. (**A**) Representative Western blot image and (**B**) quantification of densitometries of relative bands. Data are expressed as mean ± SD of fold induction relative to β-actin of three independent experiments (*n* = 3). ** *p* < 0.01 vs. CTR.

**Figure 4 ijms-23-01144-f004:**
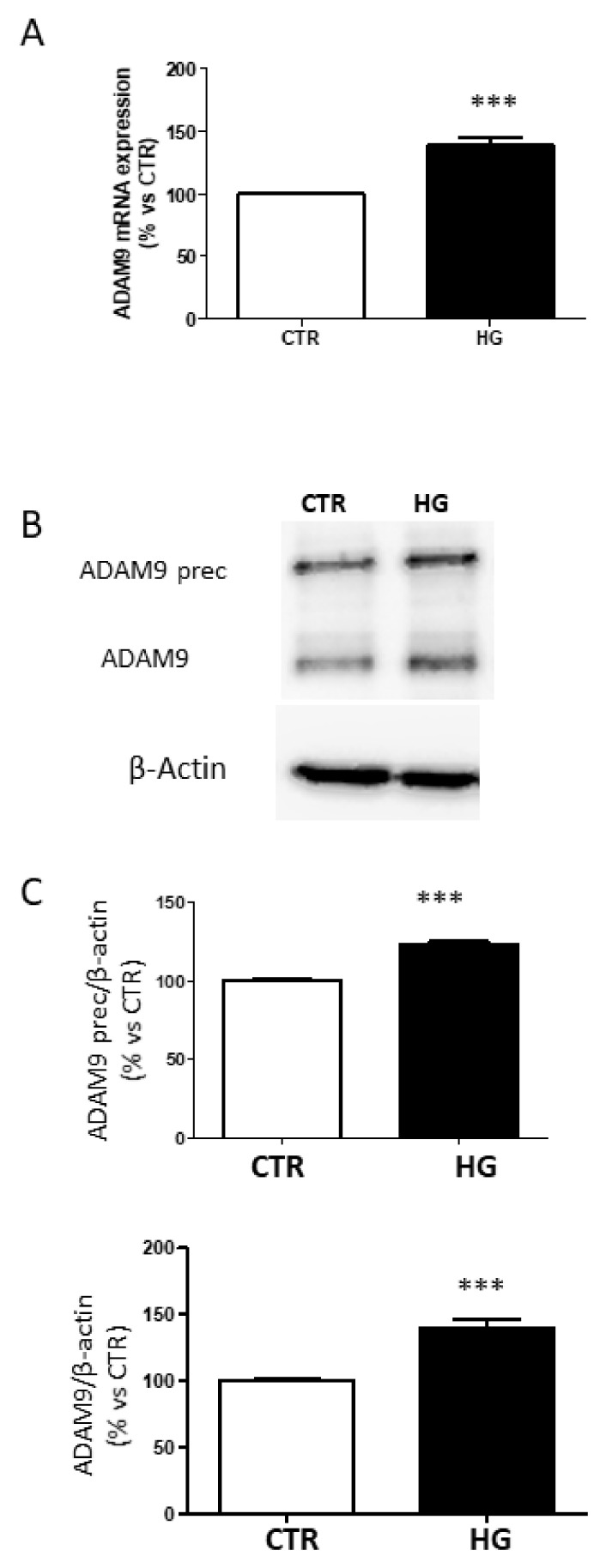
Expression of ADAM9 in ARPE-19 cells exposed to HG. (**A**) mRNA levels of ADAM9. (**B**) Representative image of ADAM9 protein expression. (**C**) Quantification of densitometries of Western blot bands. Data are expressed as mean ± SD of fold induction relative to β-Actin (*n* = 3). *** *p* < 0.001 vs. CTR.

**Figure 5 ijms-23-01144-f005:**
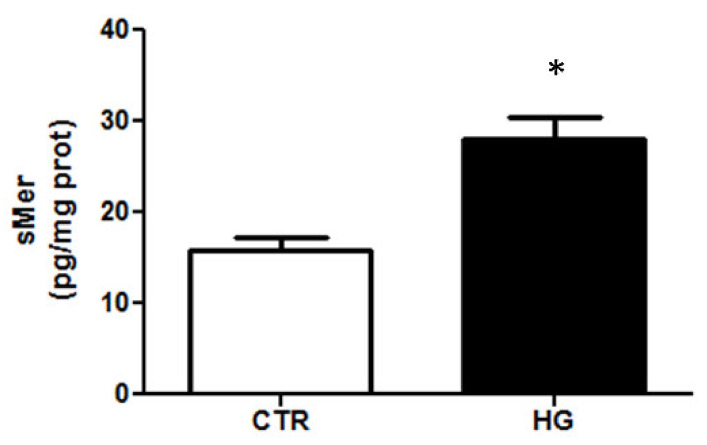
sMer measured by ELISA. Data are presented as the mean ± SD of three experiments (*n* = 3). * *p* < 0.05 vs. CTR.

**Figure 6 ijms-23-01144-f006:**
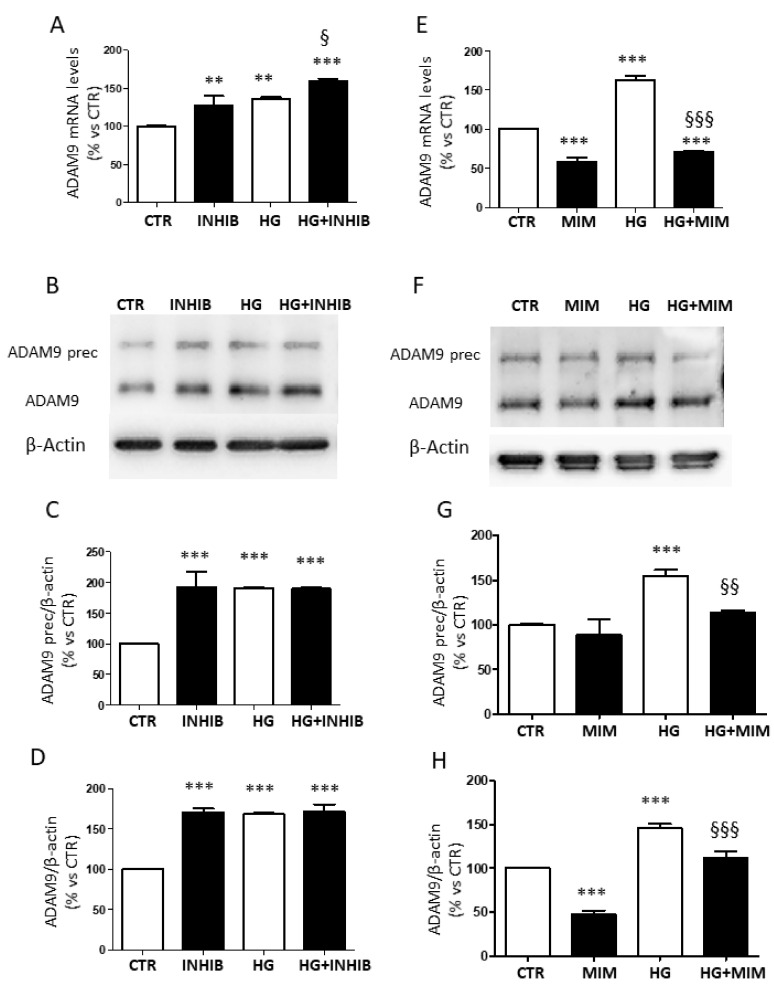
Expression of ADAM9 in ARPE-19 cells cultured for 24 h in standard medium (CTR) and in presence of high glucose (HG), silencing (INHIB), and upregulating (MIM) miR-126 under control conditions. (**A**,**E**) ADAM9 gene expression normalized vs. housekeeping genes. (**B**,**F**) Representative Western blot of ADAM9 protein expression with quantification of densitometries of both precursor (**C**,**G**) and active form (**D**,**H**). Data are expressed as mean ± SD of fold induction relative to β-actin of three independent experiments (*n* = 3). ** *p* < 0.01 and *** *p* < 0.001 vs. CTR; ^§^ *p* < 0.05 HG + INHIB vs. HG; ^§§^ *p* < 0.01 and ^§§§^ *p* < 0.001 HG + MIM vs. HG.

**Figure 7 ijms-23-01144-f007:**
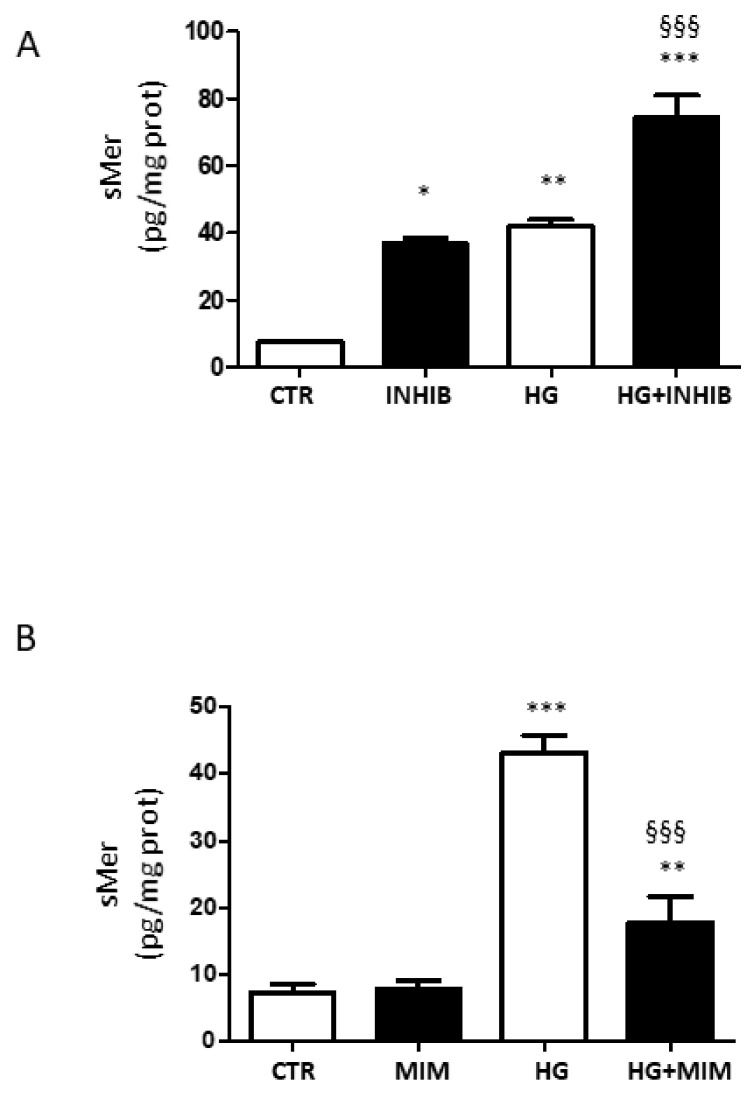
sMer measured by ELISA in ARPE-19 cells cultured for 24 h in standard medium (CTR) and in the presence of high glucose (HG), (**A**) silencing (INHIB), and (**B**) upregulating (MIM) miR-126. Data are presented as the mean ± SD of three experiments (*n* = 3). * *p* < 0.05, ** *p* < 0.01, and *** *p* < 0.001 vs. CTR; ^§§§^ *p* < 0.001 HG + MIM vs. HG.

**Figure 8 ijms-23-01144-f008:**
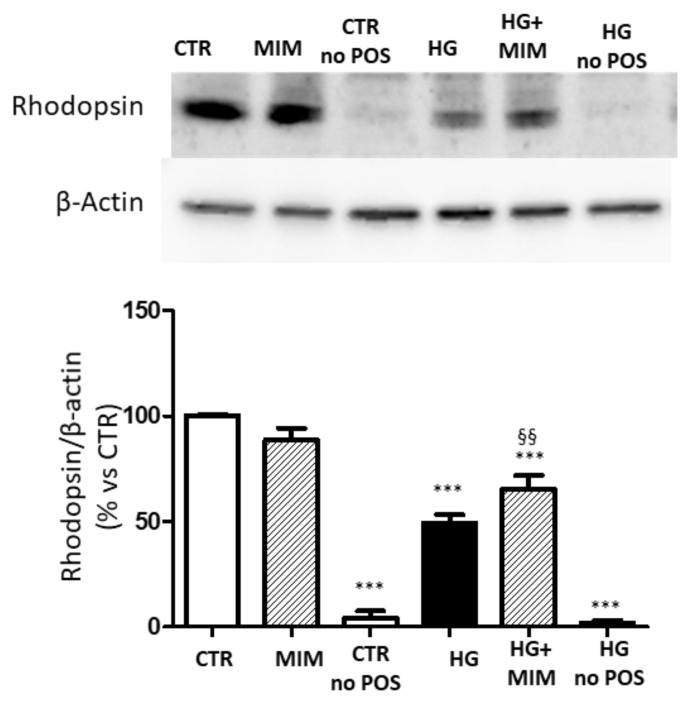
Detection of rhodopsin in ARPE-19 cells after phagocytosis assay. Cells were transfected with mir-126 mimic and then incubated with POS for 4 h. Lysates from cells labeled CTR no POS and HG no POS did not receive POS and are negative controls. (**A**) Representative Western blot image and (**B**) quantification of densitometries of rhodopsin. Data are expressed as mean ± SD of fold induction relative to β-actin of three independent experiments (*n* = 3). *** *p* < 0.01 vs. CTR; ^§§^ *p* < 0.01 vs. HG.

**Figure 9 ijms-23-01144-f009:**
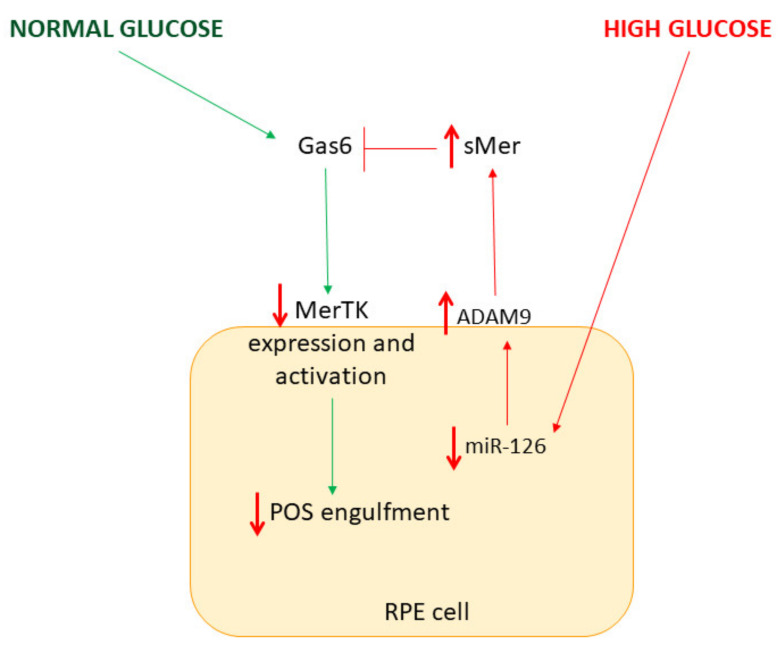
Schematic representation of HG-impaired MerTK signaling in ARPE-19 cells.

## Data Availability

The data presented in this study are available on request from the corresponding author.

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
