# Peer review of "High Glucose Impairs Expression and Activation of MerTK in ARPE-19 Cells"

_ijms, 2022, doi:10.3390/ijms23031144_

Round 1

Reviewer 1 Report

This paper reports the effects of hyperglycaemia in regulation of photoreceptors (POS) phagocytosis by alterations in expression and activation of MerTK.

The objectives of the study are well defined, and the introduction provides a state of art with appropriate references. Nevertheless, the results and the discussion have to be improved, taking in account the reviewer´s comments/suggestions and questions in the attached marked manuscript.

Author Response

This paper reports the effects of hyperglycaemia in regulation of photoreceptors (POS) phagocytosis by alterations in expression and activation of MerTK.

The objectives of the study are well defined, and the introduction provides a state of art with appropriate references. Nevertheless, the results and the discussion have to be improved, taking in account the reviewer´s comments/suggestions and questions in the attached marked manuscript.

We thank you for the comments. We revised the manuscript accordingly.

Lines 9-10: We added full name for POS and RPE: “….of photoreceptor outer segments (POS) in Retinal Pigment Epithelial (RPE) cells.”

Line 69 (now line 80): We added the missing capture (Figure 1B and 1C)

Line 72 (now line 99): We corrected the sentence: “(B) Representative Western blot analysis of MerTK protein expression and (C) quantification of densitometries of relative bands

Line 88 (now line 121): This sentence is not clear.

Reply: We modified the sentence to make it more comprehensible:  “Gas6 interacts with the extracellular region of MerTK, leading to receptor activation”.

Line 104 (now line 147): (Figure 4B and 4C) Figure 4C must be mentioned in the text!

Reply: We added the missing capture.

 Lines 128, 130 (now lines 173-176): This Fig 6 D is not labeled in fig, This Fig 6 H is not labeled in fig. Line 132: This figure and the legend are very confusing....- G and H are missing, - The x labels of grafics are confusing. Maybe you should label it like in figures B and E: CTR, CTR+INHIB (or MIM); HG; HG+INHIB (or MIM). Otherwise it is difficult to understand the difference between the two INHIB columns.

Reply: We apologize for our mistakes, we modified Figure 6 adding G and H, and correcting the x labels.

Line 142 (now line 204): why didn't test for miR-126 inhibition?

Line 195 (now line 263): The inhibition of miR-126 increased sMer levels? Results just show what happens in sMer levels when mR-126 is over expressed.

Reply: We apologized for the missing data. We analysed sMer levels also in ARPE-19 cells treated with Mir-126 inhibitor to verify the involvement of ADAM9 in sMer production. We found that treatment with miR-126 inhibitor further increased levels of sMer in ARPE-19 cells exposed to HG, confirming the role of ADAM9 in MerTK shedding. We modified Figure 7 and the text by adding these results (line 192): “To verify whether miR-126 affects the activity of ADAM9, we evaluated levels of sMer in ARPE-19 cells transfected with miR-126 inhibitor or mimic. Transfection of ARPE-19 cells with miR-126 inhibitor significantly increased levels of sMer (Figure 7A). Overexpression of miR-126 counteracted the rise of sMer in cells exposed to HG (Figure 7B).”

Line 189: How do you explain that a upregulation of miR-126 in CTR leads to a downregulation of ADAM9 but then it doesn´t lead to downregulation of sMer?

Reply: Here we found that treatment with miR-126 mimic down-regulates, but does not abrogate, production of ADAM9, suggesting residual activity of this enzyme. Levels of sMer detected in the medium collected from CTR cells are so low, that, probably, we could not detect significant reduction of sMer levels when ADAM9 was down-regulated. We added this comment in the discussion (line 260): “Here we found that inhibition or overexpression of miR-126 leads respectively to up-regulation or down-regulation of ADAM9 expression, demonstrating that ADAM9 is a target of miR-126 in ARPE-19 cells. Contextually, we observed that inhibition of miR-126 increased the levels of sMer in ARPE-19 cells, whereas miR-126 overexpression strongly reduced the amount of sMer in media from cells cultured in diabetic con-dition. Lack of significant reduction in sMer levels in ARPE-19 cells treated with miR-126 mimic under normal glucose condition may be due to the low levels of sMer found in medium collected from CTR cells. Our data demonstrate that the rise in sMer detection is correlated with the increment of ADAM9 expression, confirming that ADAM9 is involved in MerTK shedding.  …”

Line 216: How do you explain the normal rhodopsin values found in CTR when miR-126 is overexpressed?

Reply: It has been reported that POS phagocytosis is regulated by MerTK expression. In this study, all experiments are performed in an in vitro model of phagocytosis, therefore, the amount of POS engulfed by CTR cells represents the maximum amount that this system can process. Our data showed that levels of sMer in medium from cells treated with miR-126 mimic are comparable to those of CTR cells (Fig. 7), therefore, the numbers of MerTK receptors that can engulf POS are comparable too. Consistently, we found normal levels of Rhodopsin in CTR when miR-126 was overexpressed.

Reviewer 2 Report

This study investigated the effects of glucose on the expression and activation of MerTK 2 in ARPE-19 cells, which was interesting. The authors did a lot of work and the results were clearly presented. I have some comments for your reference.

  1. The results in this study were obtained based on in vitro ARPE-19 cells. Thus, the hyperglycaemia in the title was not reasonable, and the reviewer highly recommends that “hyperglycaemia” should be changed to “high glucose”.
  2. Please give a full name for MerTK and RPE in the section of Abstract.
  3. Please indicate the novelty and significance of the study in the last part of the Introduction.
  4. Why did the authors use glucose treatment for 24 h? Did you have any data for other time points, such as 6 h, 12 h, or 48 h? The concentration of glucose in the medium should be labeled, and the concentration of high glucose used in ARPE-19 cells should be also clarified.
  5. The sequence of primers used in the study should be added.
  6. The dilution fold for the primary and secondary antibodies should be added.
  7. How about the effect of high glucose on the protein expression of the phosphorylated MerTK under the condition without Gas6 treatment? For Fig.2, why the blots of phMerTK were different at the time point of zero between CTR and HG groups?
  8. It would be very helpful if authors could add a summarized figure to clearly show the signaling pathway by which high glucose impaired expression and activation of MerTK 2 in ARPE-19 cells.
  9. Did the authors have any in vivo data for supporting the conclusion obtained in this study?

Author Response

This study investigated the effects of glucose on the expression and activation of MerTK 2 in ARPE-19 cells, which was interesting. The authors did a lot of work and the results were clearly presented. I have some comments for your reference.

We thank you for the comments. We revised the manuscript accordingly.

  1. The results in this study were obtained based on in vitro ARPE-19 cells. Thus, the hyperglycaemia in the title was not reasonable, and the reviewer highly recommends that “hyperglycaemia” should be changed to “high glucose”

Reply: We replaced “hyperglycaemia” with “high glucose”.

2. Please give a full name for MerTK and RPE in the section of Abstract.

Reply: We added full name for MerTK and RPE in the section of Abstract.

3. Please indicate the novelty and significance of the study in the last part of the Introduction.

Reply: We modified the last part of Introduction, indicating the novelty and significance of the study: “RPE dysfunction is responsible for several diabetic ocular diseases, often related to retinal blood barrier breakdown. In addition, phagocytosis of photoreceptors is impaired in several pathologies, including diabetes [2,12,13]. However, molecular mechanisms underlying the dysfunctional POS phagocytosis in HG conditions remain to be elucidated. We previously demonstrated that exposure of the RPE cell line ARPE-19 to HG significantly decreased miR-126 levels [14]. In this study, we show that HG impairs the expression and activation of MerTK in ARPE-19 cells, leading to defective phagocytosis. Moreover, we found that overexpression of miR-126 can reverse HG damage by down-regulating ADAM9 production.

4. Why did the authors use glucose treatment for 24 h? Did you have any data for other time points, such as 6 h, 12 h, or 48 h?

Reply: We have chosen to treat cells with HG for 24 h, because this time-exposure allows to observe significant HG-induced alteration in both mRNA levels and protein expression. A shorter time-exposure, such as 6 or 12 h, may prevent to detect modification in protein transduction, whereas longer time-exposure may prevent to detect modification in mRNA levels (Sanguineti R, Puddu A, Nicolò M, Traverso CE, Cordera R, Viviani GL, Maggi D. miR-126 Mimic Counteracts the Increased Secretion of VEGF-A Induced by High Glucose in ARPE-19 Cells. J Diabetes Res. 2021 Feb 24;2021:6649222. doi: 10.1155/2021/6649222. PMID: 33709000; PMCID: PMC7932804; Xiao H, Liu Z. Effects of microRNA‑217 on high glucose‑induced inflammation and apoptosis of human retinal pigment epithelial cells (ARPE‑19) and its underlying mechanism. Mol Med Rep. 2019 Dec;20(6):5125-5133. doi: 10.3892/mmr.2019.10778. Epub 2019 Oct 30. PMID: 31702814; PMCID: PMC6854520.; Maugeri G, Bucolo C, Drago F, Rossi S, Di Rosa M, Imbesi R, D'Agata V, Giunta S. Attenuation of High Glucose-Induced Damage in RPE Cells through p38 MAPK Signaling Pathway Inhibition. Front Pharmacol. 2021 May 7;12:684680. doi: 10.3389/fphar.2021.684680. PMID: 34025440; PMCID: PMC8138305.)

5. The concentration of glucose in the medium should be labeled, and the concentration of high glucose used in ARPE-19 cells should be also clarified.

Reply: We used respectively 1 and 4.5 g/L glucose in CTR and HG medium. We added this information in Section 4.1 “Before each experiment, cells were washed with phosphate buffered saline (PBS) (Eu-roclone, Milan, Italy) and cultured in normal glucose condition (1 g/L glucose, CTR). Once reached confluence, cells were cultured in control medium (CTR) or high glucose (4.5 g/L glucose, HG) for 24 hours and processed for each analysis.”

6. The sequence of primers used in the study should be added.

Reply: We added this information in Section 4.6: “…Once the cells reached 70% confluence, they were transfected with has-miR-126-3p miRCURY LNA Power inhibitor (miR-126 inhib: GCATTATTACTCACGGTACG), has-miR-126-3p miRCURY LNA Mimic (miR-126 mimic: UCGUACCGUGAGU-AAUAAUGCG) and the respective scramble negative controls (inhib: TAACACGTC-TATACGCCCA; mimic: GAUGGCAUUCGAUCAGUUCUA all from Ex-iqon-Qiagen,Milan, Italy) using jetPRIME transfection reagent (Polyplus-transfection, New York, USA)….” Sequence of primers used in quantitative RT-PCR are indicated in section 4.2 by the assay ID (for instance, Hs01031968_m1).

7. The dilution fold for the primary and secondary antibodies should be added.

Reply: We added this information in section 4.3: “…Filters were blocked in Protein Free T20 Blocking Buffer (Pierce Biotechnology, Rock-ford, IL, USA) and incubated overnight at 4°C with the following primary specific an-tibodies diluted 1:1000: Mer (D21F11, cat. 4319), Mer (348E6, cat. 9178) and ADAM9 (cat. 2099) from Cell Signaling Technology, Beverly, MA, USA; anti-MerTK (phospho Y749) +TYRO3 (phospho Y681) antibody cat. ab192649 from abcam, Cambridge, UK. Secondary specific horseradish peroxidase-linked antibodies, diluted 1:3000, were added for 1 h at room temperature. Bound antibodies were detected using the en-hanced chemiluminescence lighting system (LiteAblot EXTEND, Euroclone), according to the manufacturer’s instructions. Each membrane was stripped (Restore PLUS West-ern Blot Stripping Buffer, Pierce Biotechnology, Rockford, IL, USA) and probed for β-actin (1:3000, Cell Signaling Technology, Beverly, MA, USA)…”

8. How about the effect of high glucose on the protein expression of the phosphorylated MerTK under the condition without Gas6 treatment? For Fig.2, why the blots of phMerTK were different at the time point of zero between CTR and HG groups?

Reply: Actually, HG did not affect protein expression of phMerTK (see Fig. 2B). The image in Figure 2 may be misleading. Indeed, there is no significant difference between levels of phMerTK in CTR and HG-treated cells, when results are reported to relative β-actin. We corrected the label in Fig. 2B.

9. It would be very helpful if authors could add a summarized figure to clearly show the signaling pathway by which high glucose impaired expression and activation of MerTK 2 in ARPE-19 cells.

Reply: We added a summarized figure at the end of discussion. “Mechanisms through which hyperglycemia affects POS phagocytosis are not yet understood. Our study suggests that the HG-induced decreased levels of miR-126 leads to up-regulation of ADAM9 expression, and this, in turn, increases cleavage of MerTK. The consequent rise in sMer coupled to reduced expression of MerTK may impair activation of intracellular signalling that leads to binding and internalization of POS (Figure 9).

10. Did the authors have any in vivo data for supporting the conclusion obtained in this study?

Reply: We thank the Reviewer for this suggestion, but, we performed only in vitro experiments.

Round 2

Reviewer 1 Report

The authors made the corrections/changes suggested by the reviewer and the manuscript was improved, so I believe it can be accepted for publication in the IJMS.